# Occurrence of Aflatoxin M1 over Three Years in Raw Milk from Croatia: Exposure Assessment and Risk Characterization in Consumers of Different Ages and Genders

**DOI:** 10.3390/foods14132396

**Published:** 2025-07-07

**Authors:** Nina Bilandžić, Ines Varga, Bruno Čalopek, Božica Solomun Kolanović, Ivana Varenina, Maja Đokić, Marija Sedak, Luka Cvetnić, Damir Pavliček, Ana Končurat

**Affiliations:** 1Laboratory for Residue Control, Department of Veterinary Public Health, Croatian Veterinary Institute, Savska cesta 143, 10000 Zagreb, Croatia; varga@veinst.hr (I.V.); calopek@veinst.hr (B.Č.); solomon@veinst.hr (B.S.K.); varenina@veinst.hr (I.V.); dokic@veinst.hr (M.Đ.); sedak@veinst.hr (M.S.); 2Laboratory for Mastitis and Raw Milk Quality, Department for Bacteriology and Parasitology, Croatian Veterinary Institute, Savska cesta 143, 10000 Zagreb, Croatia; lcvetnic@veinst.hr; 3Croatian Veterinary Institute, Veterinary Institute Križevci, Zakmardijeva 10, 48260 Križevci, Croatia; pavlicek.vzk@veinst.hr (D.P.); koncurat.vzk@veinst.hr (A.K.)

**Keywords:** aflatoxin M1, cow milk, occurrence, estimated daily intakes, risk assessment, Croatia

## Abstract

In this study, the frequency of aflatoxin M1 (AFM1) occurrence in raw milk was investigated across different seasons over a three-year period from 2022 to 2024 in Croatia. Risk assessment was conducted using estimated daily intake (EDI), hazard index (HI), and margin of exposure (MOE) for various age groups and both genders. The frequency of AFM1 detection above the maximum level (ML) ranged from 1.60% to 15.1%. The average incidence of AFM1 exceeding the ML was 5.67%, with the highest incidence recorded in autumn 2024. AFM1 levels within the limit of detection (LOD) and ML were found in 13% of the samples. The average mean value of AFM1 over the three-year period was 19.2 ng/kg. The highest mean AFM1 EDI values were determined for toddlers (0.61–0.67 ng/kg bw/day) and children (0.41–0.43 ng/kg bw/day). The lowest EDI values were observed in elderly females and males (0.058–0.074 ng/kg bw/day). The EDI values for females and males were slightly different. The risk assessment, based on the HI and MOE, indicated that toddlers and children are at the highest risk of exposure to AFM1, which raises significant health concerns. Additionally, consumers of large quantities of milk face a high risk of exposure, particularly during the spring and autumn seasons. For adults and the elderly, milk consumption does not pose a serious health risk.

## 1. Introduction

Mycotoxins represent a significant challenge in the realm of food safety today, as they contaminate a wide variety of food products and can lead to adverse health effects across all age groups [1]. Among the various mycotoxins, aflatoxins are prevalent in numerous food items, ranging from cereals to milk and dairy products [2,3]. These compounds are secondary metabolites produced by the strains *Aspergillus flavus* and *A. parasiticus*, with aflatoxin B1 (AFB1) being the most notable. Upon ingestion by a lactating mammal, AFB1 is metabolized in the liver to the hydrolyzed metabolite aflatoxin M1 (AFM1), which is subsequently excreted in milk [4].

The presence of AFM1 in milk and dairy products poses a significant public health concern, as individuals of all age groups consume milk daily, particularly children and infants, who are estimated to have the highest levels of exposure [2,5,6]. The processes employed in milk processing, such as high-temperature treatment, pasteurization, or sterilization, are ineffective in deactivating or degrading AFM1 [7]. Consequently, the risk of exposure to this contaminant can only be mitigated through continuous monitoring and regulation of the maximum level (ML) of AFM1 in milk, which in the European Union is set to exceed 50 ng/kg [8]. AFM1 is carcinogenic when ingested orally and exhibits a carcinogenic potency that is one-tenth that of AFB1 [9]. The International Agency for Research on Cancer (IARC) has classified both AFB1 and AFM1 as Group 1 human carcinogens based on findings from animal and epidemiological studies [10].

The production of aflatoxin by *Aspergillus flavus* depends on various factors, including temperature, water activity, and carbon dioxide levels [11,12]. It is common for hot and humid climates to promote the proliferation of mycotoxigenic fungi, leading to the production of aflatoxins [13]. The occurrence of aflatoxins is particularly prevalent in crops subjected to abiotic stress, such as high temperatures or drought, which is characteristic of tropical and subtropical regions, including Africa, Central America, and South or Southeast Asia [1,13,14,15,16,17].

In the earlier decades of the 20th century, aflatoxin contamination was not typical in European countries. However, due to climate change affecting the Mediterranean, Southeastern, and Central European regions, evidenced by increased frequency and intensity of extreme temperatures, prolonged droughts, heat waves, elevated winter and summer temperatures and precipitation, and changes in precipitation amounts and patterns, the incidence of aflatoxins has risen significantly [18,19,20,21]. These climatic conditions over the past two decades have contributed to a notable increase in AFB1 levels in animal feed [22,23]. Consequently, high concentrations of AFM1 in milk have been recorded in Southeastern Europe, particularly in Croatia [24,25,26,27] and Serbia [28,29,30]. AFM1 levels fluctuate across various seasons and years of observation. High frequencies of AFM1 occurrence exceeding the ML were recorded during the winter and spring months of 2013 in Croatia [24], while lower frequencies were observed during the five-year period from winter 2016 to winter 2022 [21]. A study conducted in Serbia from 2015 to 2018 showed variations in AFM1 levels, with high frequencies of occurrence noted in winter and autumn [30].

Increasingly extreme climatic conditions critically impact crop phenology, as well as the growth of fungi and the production of mycotoxins, thereby influencing food safety [12]. Consequently, constant monitoring of the frequency of AFM1 occurrence in raw milk and the determination of its concentrations are essential for assessing human exposure and evaluating the risks to human health over prolonged periods, particularly at low doses [31]. Therefore, the risks associated with exposure to AFM1, as well as the characterization of these risks across different age groups and genders, have been examined in various studies. These studies focus on calculating the Estimated Daily Intake (EDI) and the Hazard Index (HI) [2,6,14,18,21]. Furthermore, for genotoxic and carcinogenic compounds such as aflatoxins, risk assessment using the Margin of Exposure (MOE) approach has been recommended [5] and implemented [1,3,23].

In Croatia, a continuous strategy for controlling AFM1 in milk has been implemented, and this study serves as a continuation of the research on the occurrence and concentrations of AFM1 in milk from primary production. Additionally, a more comprehensive exposure and risk assessment was conducted for all age groups within the Croatian population, whereas previous studies focused solely on the adult population.

## 2. Materials and Methods

### 2.1. Sample Collection

Raw milk samples were collected from spring 2022 to winter 2024. A total of 1951 milk samples were obtained from dairy farms and milk processing plants across all geographical regions of Croatia. The total number of samples collected during the spring, summer, autumn, and winter seasons over three years is as follows: in 2022: 151, 175, 269, and 250; in 2023: 87, 88, 164, and 114; in 2024: 120, 125, 278, and 49. Additionally, the milk samples were categorized by sampling season. For the collection of milk samples, sterile 0.5 L plastic bottles were utilized. During transport to the laboratory, the samples were maintained at a temperature of 2–8 °C and were subsequently stored at −20 °C until analysis was conducted.

Samples were thawed and centrifuged for 10 min at 3500× at 10 °C prior to analysis. The upper cream layer was removed by aspirating aspiration using a Pasteur pipette. Skimmed milk was then used directly in the test (100 µL per well).

### 2.2. AFM1 Quantification

For the detection of AFM1, the ELISA method was employed using the R-Biopharm AFM1 test kit (R1121, R-Biopharm AG, Darmstadt, Germany). The AFM1 standard was sourced from Sigma-Aldrich (St. Louis, MO, USA), while the internal standard for AFB1 was obtained from a different laboratory.

Details regarding the instrumentation used for the preparation of milk samples and the determination of optical density were described in a previous study [24]. The procedures for preparing standard stock solutions, working solutions, the contents of reagents within the kit, and subsequent procedures for AFM1 analysis were conducted according to the instructions provided in the ELISA test kit, as previously described [24]. The method was validated in accordance with Commission Decision 2002/657/EC [32]. The validation parameters included a limit of detection (LOD) of 22.2 ng/kg, a limit of quantification (LOQ) of 34.2 ng/kg, and a recovery rate ranging from 87.4% to 107.5% (spiked levels of 25, 50, and 75 ng/kg), with coefficients of variation between 8.7% and 17.0%.

AFM1 concentrations were detected using the ELISA method. In cases where AFM1 concentrations exceeded 50 ng/kg, samples were reanalyzed using the confirmation method of ultra-high-performance liquid chromatography with tandem mass spectrometry (UHPLC-MS/MS). This analysis was conducted with the UHPLC 1290 Infinity II system and the Triple Quad LC/MS 6470A mass spectrometer (Agilent, Palo Alto, CA, USA). The extraction and clean-up procedures consisted of centrifugation and purification using immunoaffinity columns (IAC VICAM Afla M1™ HPLC, VICAM, Milford, MA, USA), as described in a previous study [22]. The chemicals, reagents, and details regarding the instrumentation used for the preparation of milk samples were also presented in detail in the earlier publication [22].

Chromatographic separation was achieved using a Zorbax Eclipse Plus C18 Rapid Resolution HD column, measuring 3.0 × 50 mm with a particle size of 1.8 µm (Agilent Technologies, Palo Alto, CA, USA). The chromatographic and operational conditions of the UHPLC-MS/MS system were detailed in a previous study [22]. The method was validated according to the criteria established by Commission Decision 2002/657/EC [32], with the following validation parameters: LOD of 2.8 ng/kg, LOQ of 11.0 ng/kg, and recovery rates ranging from 100.6% to 102.1% at concentration levels between 0.1 and 0.75 µg/kg. The relative standard deviations (RSD, %) for intra-laboratory reproducibility were lower than 18.3%. The validation results obtained meet the limits set by Commission Regulation 401/2006 [33].

### 2.3. Dietary Exposure and Risk Characterization

For the assessment of risk exposure to milk consumption across different population age groups, estimated daily intakes (EDI) and hazard index (HI) were utilized. The estimation of daily intakes (EDI, ng/kg bw/day) of AFM1 by consumption was calculated using the equation EDI = C × MS, where C represents the mean concentration or the 95th percentile (P95) of AFM1 (ng/kg), and MS denotes the meal size, which is the amount of milk consumed per day per body weight (g/kg body weight/day) [21]. To calculate the EDI value, the total mean values from the results of all three years of monitoring AFM1 concentrations, ranging from the LOD to 50 ng/kg, as well as those exceeding 50 ng/kg, were employed.

Milk and dairy product consumption among the Croatian population, for both sexes, is documented in the EFSA Comprehensive European Food Consumption Database [34]. Chronic consumption of milk and dairy products (g/kg body weight/day) is reported as both the mean and P95 for various age groups, including toddlers, children, adolescents, adults, the elderly, and the very elderly (values are presented in Section 3.3). The consumption of large quantities of milk and dairy products is represented by the P95 values to evaluate the worst-case scenarios.

The evaluation of chronic risk toxicity associated with milk consumption was conducted by calculating the hazard index (HI) using the equation: HI = EDI/Tolerable Daily Intake (TDI). The TDI value utilized in risk assessment studies [3,21,35,36,37,38] is based on Kuiper-Goodman [39], which specifies a value of 0.2 ng/kg bw/day. No risk or adverse effects on consumer health are presumed when the HI is less than 1. An HI greater than 1 indicates a potential toxic health effect.

Risk characterization was also conducted using the MOE which serves as an index for assessing the risk of oral exposure to carcinogenic and genotoxic substances, such as AFM1 [5]. The European Food Safety Authority (EFSA) establishes the MOE calculation as follows: MOE = BMDL/EDI, where BMDL refers to the benchmark dose level confidence limit of 10% (BMDL10). The BMDL_10_ ratio was established at 400 ng/kg body weight per day for the induction of hepatocellular carcinomas (HCC) in male rats following exposure to AFB1, but in the absence of a specific BMDL10 for AFM1 a potency factor of 0.1 relative to AFB1 was utilized. For MOE values equal to or greater than 10,000, it can be concluded that milk AFM1 poses minimal concern regarding public health. However, when the MOE is less than 10,000, it presents a significant risk of HCC for consumers [1,3,5,37].

### 2.4. Statistical Analysis

AFM1 concentrations were calculated as the mean ± standard deviation (SD), 95th percentile, along with the minimum and maximum values. Concentrations below the LOD were set to half of the LOD values. Statistical analyses were conducted using Stata version 13.1 for Windows (64-bit x86-64) (StataCorp LP, College Station, TX, USA).

## 3. Results and Discussion

### 3.1. AFM1 Concentration Levels

The mean concentrations of AFM1 were determined for the total number of samples collected per season over three years, as well as for two concentration ranges: between the LOD and the ML, and above the ML. These values, along with the range of positive concentrations observed during the three seasons over the years 2022, 2023, and 2024, are reported in Table 1. The total mean concentrations across the seasons over three years ranged from 11.1 to 28.9 ng/kg, with an overall mean value of 19.2 ng/kg. For the concentrations between the LOD and ML, mean values varied from a minimum of 27.6 ng/kg (summer 2022) to a maximum of 40.9 ng/kg (summer 2024), resulting in an overall mean of 30.2 ng/kg. The mean values of AFM1 for samples above the ML ranged from 69.9 ng/kg to 230.0 ng/kg, with the highest value recorded in spring 2022. The lowest overall mean value of 82.8 ng/kg was observed in 2023, while the highest mean of 129.9 ng/kg was noted in 2022. The maximum values measured ranged from 92.0 to 1250 ng/kg, with the highest value measured in the winter of 2022. The overall mean value for AFM1 exceeding ML of all three years of measurement is 108.5 ng/kg. The average mean value of all results over the three-year period was 17.3 ng/kg.

Similar values were observed during the AFM1 crisis in Croatia in 2013, with average concentrations of positive samples ranging from 64.2 to 131 ng/kg. Maximum values of 1105 ng/kg and 1135 ng/kg were recorded in February and April, respectively [24], and a concentration of 764.4 ng/kg was noted during the winter of 2013 [26]. In the spring and autumn of 2016, lower mean levels above the ML were determined, ranging from 52.1 to 84.5 ng/kg [27]. A study conducted over a five-year period from winter 2016 to winter 2022 in Croatia revealed a similar range of concentrations, from 50.7 ng/kg to a peak value of 1110 ng/kg in autumn 2021 [21]. However, the highest mean levels of AFM1 exceeding the ML of 387.8 ng/kg and 209.8 ng/kg were recorded in autumn 2021 and winter 2021/2022, respectively.

The highest concentrations of AFM1 measured in neighboring Serbia were 220 ng/kg during the autumn of 2015–2018 [30] and 554 ng/kg during the period from 2015 to 2022 [40]. The mean values of AFM1 exceeding the ML ranged from 125 to 660 ng/kg [30]. Furthermore, the highest total mean values reported in Serbia were 20.77 ng/kg for the period 2017–2019 and 32.9 ng/kg for 2015 to 2022, which are higher than those observed in this study [40,41]. The lowest maximum and mean values of AFM1, recorded at 58 ng/kg and 10.94 ng/kg in raw milk in Italy, were reported during the period from 2013 to 2019 [6]. Furthermore, a separate study conducted in Italy from 2014 to 2020 found maximum concentrations of positive samples of cow’s milk ranging from 68 to 146 ng/kg [42].

High levels of AFM1 are characteristic of tropical or subtropical climates, which are defined by extremely high temperatures and significant rainfall accompanied by high relative humidity [37]. Recent studies conducted in various countries within these regions have reported elevated maximum values of AFM1: 334 ng/kg in Armenia [43], 2000 ng/kg in Ethiopia [42], 750 ng/kg in Ecuador [44], 3520 ng/kg in Ghana [1], and 2913 ng/kg in India [45]. The total mean AFM1 values in milk from Iran were 77 ng/kg and 84 ng/kg, as reported by Hasninia et al. [46] and Massahi et al. [37]. In Ethiopia and Ghana, the total mean values varied by region, ranging from 40 to 550 ng/kg [38] and from 260 to 860 ng/kg [1].

### 3.2. Seasonal Frequency of AFM1 Occurrence

A total of 1951 samples of raw cow’s milk collected in Croatia over the three years 2022, 2023, and 2024 were analyzed for AFM1 concentrations. In Figure 1, the frequencies of AFM1 detection in milk samples are presented for each annual period over the three-year span. The frequencies of milk detection were monitored across three concentration ranges: below the LOD, in the range of the LOD (22.2 ng/kg) and the maximum level (ML) defined by the European Union [8], and above the ML (>50 ng/kg).

During 2022, the incidence of concentrations exceeding the LOD to the ML ranged from 0.66% in spring to a peak of 14.9% in autumn. The incidence of positive samples above the MRL varied from 2.65% to a maximum of 7.43%, also recorded in autumn. In 2023, a higher incidence of concentrations above the LOD was observed, with values of 35.6% in winter and 33% in spring. The percentage of positive samples during the seasons ranged from 3.05% in summer to 6.81% in spring. In winter and spring 2024, no occurrences of AFM1 above the LOD were detected, and during winter, there were also no values exceeding the MRL. In spring and summer, the proportion of positive samples was very low, at just 1.6% of the total. However, the highest incidence of AFM1 above the MRL, at 15.1%, was recorded in autumn. Additionally, in December (winter), the highest percentage of concentrations above the LOD, at 44.9%, was noted. Concentrations below the LOD over the three years ranged from 51.7% to 100%. Overall, during these three years, the average incidence of AFM1 above the MRL was 5.67%, with concentrations above the LOD occurring in 13% of the samples. In total, AFM1 below the LOD was detected in 81.3% of the samples.

The frequencies of AFM1 occurrence exceeding the ML determined in this study are significantly lower than those reported in 2013, during the AFM1 crisis. In February, March, and April of that year, the frequencies were 45.9%, 35.4%, and 29.9%, respectively [24]. Additionally, in the autumn of 2013, the frequency was recorded at 9.32% [26]. On the other hand, observed incidence of AFM1 exceeding the LOD or the ML is significantly higher compared to previous surveys conducted in the spring and autumn of 2016, as well as during the five-year period from winter 2016 to winter 2022. The incidences reported were 0.4–0.9% and 0.3–11% [27], respectively, and 3.42% and 1.87% [21].

Over the past decade, numerous studies have been conducted in neighboring Serbia, which, along with Croatia, is part of the Southeastern region of Europe. A study by Miličević et al. [20] analyzed milk samples collected from 2015 to 2018 and found that the frequency of aflatoxins exceeding the ML ranged from 27% to 31% during this four-year period. In another study, the occurrence of AFM1 above the ML during the four-year period from 2015 to 2018 varied, ranging from 25.4% in winter to 65.4% in autumn, with an overall mean incidence of 46.2% [30]. A study conducted between 2017 and 2019 revealed that the incidence of AFM1 was above the LOD in 20% of the samples [41]. Further, the incidence of AFM1 occurrence between 2015 and 2022 was 16.4% [40]. During 2022, the presence of AFM1 in milk was documented through official controls, specifically the national residue monitoring programs of EU member states. The overall occurrence rate was 0.54%, with a total of nine results exceeding the ML in the following countries: Croatia, Bulgaria, Italy, Finland, and Greece [47].

Conversely, an extremely low incidence of AFM1 was reported in Italy from 2014 to 2020, with the frequency of positive milk samples ranging from 0.43% to 1.5% [48]. In Italy, more than 43,000 samples of raw milk from milk processing plants collected between 2013 and 2019 exhibited a frequency of AFM1 above the ML of only 0.03%, with 3% of the samples showing concentrations exceeding 30 ng/kg [6]. These commendable results, reflecting a low incidence of AFM1 (compared to an incidence of 2.52% from 2004 to 2008), are attributed to a comprehensive risk management strategy implemented in Italy after 2013. This strategy involves stringent controls on animal feed and dairy products through regional control plans. An “attention limit” of 40 ng/kg has been established for milk, which is enforced during extreme weather conditions. When concentrations exceed this limit, the samples are deemed suspicious, prompting additional checks and measures [6].

Other studies worldwide have reported incidences of AFM1 exceeding the ML of 7.14% in Armenia [43], 33.0% in Iran [37,46], 52.5% in Ghana [1], 53.9% in Ecuador [44], 24.2% in Pakistan [49], and 76% and 84.2% in India [45,50]. Most of these countries are located in tropical and subtropical regions, where climatic conditions favor the growth of fungi and the production of aflatoxin [1].

In accordance with previous research [21,27], a seasonal influence on the incidence of AFM1 is evident, with an increased frequency of determination (>LOD) during the autumn and winter months. However, an exception occurred in winter and spring 2023, when the occurrence of AFM1 in milk exceeded 30%. These periods correspond to increased feeding with concentrated feed, suggesting that feed substitutes such as hay and corn contained higher concentrations of AFB1. However, these levels were not sufficiently high to result in AFM1 values exceeding the ML in milk. In the autumn and winter of 2024, the frequency of AFM1 determinations above the LOD increased, with rates of 18.7% and 44.9%, respectively. Additionally, concentrations exceeding the ML were found in 15.1% of samples during the autumn.

A significant factor contributing to the presence of these contaminants is climate change, which has notably affected Europe, especially Southeastern Europe, over the past decade. This impact is characterized by notable temperature fluctuations, prolonged droughts during the summer months, and changes in precipitation patterns, including sporadic instances of extremely heavy rainfall [51]. These climatic extremes facilitate the occurrence and proliferation of toxic molds both before and during the harvest of cereals and corn, as documented over the last two decades [21,23,52,53]. This finding is supported by investigations into the presence of AFB1 in corn samples collected from various regions of Croatia over a four-year period, from 2018 to 2021. The studies revealed a consistent presence of AFB1, ranging from 14% to 40% [54].

All three observed years from 2022 to 2024 in Croatia were characterized by climatic extremes that deviated from the historical averages. The climatic data recorded by the Croatian Meteorological and Hydrological Service in 2022 indicated exceptionally warm weather in February, May, June, July, September, and December, with temperatures ranging from 1.3 °C to 4.9 °C above the multi-year average for the reference period of 1981–2010 [55]. Overall, 2022 was marked by winter, summer, and autumn temperatures that were 2.6 °C, 3.2 °C, and 2.2 °C above the average, respectively. Both spring and summer of 2022 experienced extreme dryness.

In 2023, temperatures were recorded as 1.1 °C to 4.4 °C warmer than the multi-year average (1981–2010) during the months of January, March, July, September, October, and December [56]. The overall seasonal assessment revealed a warmer winter, summer, and autumn by 3.1 °C, 1.3 °C, and 3.4 °C, respectively, compared to the average values. This year was notably rainy during the winter, spring, and summer months.

The European Environment Agency has announced that climate change is manifesting in Europe through a rate of warming that exceeds the global average. Over the past decade, the mean annual temperature in Europe has been between 2.12 and 2.19 °C higher than during the pre-industrial period. Following 2020, the years 2023 and 2024 have been recorded as the second and third warmest years in Europe since the beginning of instrumental records [57,58]. From June to September 2024, Europe experienced six heat waves, the most severe of which lasted 13 days and exhibited an anomaly of 9.2 °C. During this 97-day period, there were a total of 43 days characterized by heat waves. Almost every day of the summer recorded maximum temperatures above the average. Record-high temperatures predominantly impacted central, eastern, and southeastern Europe, as well as northern Scandinavia and southeastern Spain [58].

For Southeastern Europe, the analysis of climate parameters indicated that the number of heat stress days and tropical nights increased during the period from 2022 to 2024, along with the annual variability in the number of humid summer days. In 2024, Eastern Europe experienced record-high annual temperatures, while Southeastern Europe recorded its longest heatwave lasting 13 days. In Southeastern Europe, summer temperatures were 3.3 °C above the average. A total of 66 days were classified as “severe heat stress” (with a perceived temperature of 32 °C or higher), marking the highest number ever recorded and significantly exceeding the average of 29 days. Additionally, there were a record 23 tropical nights, during which temperatures did not drop below 20 °C, surpassing the average of eight nights and the previous record of 16 nights set in 2012. Additionally, Southeastern Europe faced the most severe dry conditions and the driest summer in a 12-year period, as indicated by the “drought index”. In this region, the variability in the number of wet summer days is increasing year by year [58].

### 3.3. Dietary Exposure and Risk Assessment for Different Age Groups

An assessment of population exposure to this contaminant across different age groups, along with risk characterization based on available toxicological reference values, is essential for evaluating the severity of potential adverse health effects. Ultimately, the findings from such assessments can be utilized to implement control measures that ensure food safety or to gather critical information for establishing risk management procedures and evaluating the likelihood of adverse health outcomes [18].

The estimated exposure of both sexes to AFM1 for different age categories of the population in Croatia, along with the risk assessment expressed as HI, is presented in Table 2. The EDI values were calculated based on the total mean concentration for the total number of samples collected per season over three years and the range of concentrations above the LOD to the ML, as well as for the overall mean value of all positive samples (>ML). For toddlers and children, the highest EDI values for mean AFM1 ranged from 1.46 to 2.41 ng/kg bw/day for both females and males. These values were observed in results exceeding 50 ng/kg. For all other age groups, within the three concentration groups, EDI values were below 1 ng/kg bw/day. The lowest values recorded were 0.037 ng/kg bw/day for elderly females and 0.047 ng/kg bw/day for elderly males within the total group of results.

For consumers who consume large quantities of milk, P95 values were used for the calculation, which gave significantly higher EDI values across all age groups, with toddlers exhibiting the highest values. Minor differences in the EDI of AFM1 between females and males can be attributed to variations in body weight and milk consumption. Higher EDI values were observed for females, except in the child and adolescent groups. 

Multiple higher EDIs were calculated based on the mean and P95 for AFM1 concentrations above the ML. The highest EDI values for toddlers were 10.9 ng/kg bw/day for males and 12.1 ng/kg bw/day for females, with additional values of 56.3 ng/kg bw/day for males and 63.6 ng/kg bw/day for females. EDI values based on P95 ranged from 1.34 ng/kg bw/day for elderly males to 12.7 ng/kg bw/day for female toddlers.

The EDI values for the LOD to ML concentration range across all age groups were similar to the European Food Safety Authority (EFSA) chronic dietary exposure values, which summarize data from various European countries [5]. EFSA reported the highest chronic dietary exposure to AFM1 for toddlers with mean and P95 values from 0.68 to 1.05 ng/kg bw/day, and from 3.80 to 4.85 ng/kg bw/day, respectively [5]. The EDI values for adults determined in this study corresponded to the values of 0.1 ng/kg body weight/day previously determined in Croatia [21].

In Serbia, the study on chronic dietary exposure to AFM1 in milk and dairy products showed lower EDI values for the mean and P95 in children, which were 0.336 and 1.009 ng/kg bw/day, respectively, than in the present study [40]. However, higher EDIs for mean and P95 were reported for adolescents, at 0.183 and 0.526 ng/kg bw/day, and for adults, with females showing values of 0.161 and 0.457 ng/kg bw/day, and males at 0.126 and 0.321 ng/kg bw/day [40]. Additionally, another study conducted in Serbia during the period from 2017 to 2019 reported lower EDI values for toddlers and children, ranging from 0.069 to 0.135 ng/kg BW/day and from 0.039 to 0.075 ng/kg bw/day for both sexes, compared to the findings of this study [41].

In an Italian study conducted from 2013 to 2018, the mean EDI values for adults ranged from 0.02 to 0.08 ng/kg bw/day, while for P95 they were found to be between 0.04 and 0.13 ng/kg bw/day [21]. In contrast, the mean and P95 EDI values for toddlers were lower, ranging from 0.49 to 1.62 ng/kg bw/day and 0.35 to 1.16 ng/kg bw/day, than in the present study. In another study from Italy, conducted between 2013 and 2019, revealed that the highest EDI values for infants and toddlers ranged from 0.24 to 30 ng/kg bw/day [6]. In contrast, the lowest EDI values for adults were found to be between 0.02 and 0.03 ng/kg bw/day, which are lower than those reported in this study.

The risk of exposure to AFM1 through milk consumption was evaluated using the HI obtained by dividing the EDI values by a safety threshold of 0.2 ng/kg bw/day [39]. The highest risk, based on mean EDI values, was identified for toddlers and children of both sexes with an HI above 1. However, the HI for adolescents, adults, and the elderly remained below 1. When assessing the HI for large portion sizes (P95) across the total group, the results between the LOD and ML were significantly elevated values. These values indicated a very high risk for toddlers and children, as well as for other age groups, with HI values ranging from 1.18 to 14.2. Furthermore, when evaluating the risk associated with EDI values for AFM1 above the ML, either for mean values or P95 values, extremely high HI values were revealed for all children across the two age groups, regardless of sex. However, for other age groups, the HI was above 1 in both scenarios.

The risk characterization results obtained in this study are consistent with previous research, which also identified the highest risks for toddlers [3,5,6,18]. However, unlike these earlier studies, this research also identified risks for children. This increased risk is likely attributable to their body weight in relation to their relatively high milk intake [3,6,18,40,41,48]. In African and Asian countries, where the prevalence of AFM1 is high and significantly higher EDI values have been recorded, the HI index has also been validated for adults, as demonstrated in Ethiopia [42] and in Iran for both adults and the elderly [37].

Further risk characterization of AFM1 exposure was conducted using the MOE approach, which is particularly important as it assesses the carcinogenicity of AFM1 to the liver. The results of the MOE assessment are presented in Table 3. The MOE values, determined based on the mean values of the total results, show values below 10,000 only for female toddlers (9523), while the values for males and other age groups exceed this safety limit. However, for the other two groups of results, the group between the LOD and ML and that above the ML, MOE values fall below the safe limit for both female and male toddlers (ranging from 1659 to 6557) and for children (ranging from 2614 to 9756). This indicates that these two age groups are at risk due to the consumption of raw cow milk, raising significant public health concerns.

Furthermore, the MOE assessment conducted for the EDI of P95 values revealed values below 10,000 for three groups of children: toddlers, children, and adolescents (ranging from 314.0 to 8889). In addition, for all three age groups of adults in the AFM1 score group above ML, MOE values below the safe limit were also observed. MOE values exceeding the safe limit were calculated for three adult groups, considering both the total group of results and those results that fell between LOD and the ML (ranging from 11,111 to 16,667).

The values obtained in this study are consistent with EFSA’s evaluation of MOEs for different age groups and support the conclusion that the results indicate a health concern for younger populations [5]. In an Italian study conducted between 2014 and 2020, MOE values were calculated for medium and high consumers [48]. As in this study, values below the safe limit were identified for toddlers, while for the children’s group, they were below the safe limit only for high consumers. For other age groups, MOE values exceeded 10,000. However, a recent study from China reported MOE values exceeding 10,000, even for children aged 2 to 4 years [36]. Conversely, studies conducted in several countries indicate an increased risk of exposure to AFM1 for older age groups. A study conducted in Serbia and Greece in 2018 found MOE values below the safe limit, thereby indicating an increased health risk for students aged 18 to 27 years [40]. Additionally, studies from Ghana reported MOE values below 10,000 for infants, toddlers, children, adolescents (ranging from 1600 to 5000), and adults (ranging from 2105 to 6666) [1]. Furthermore, studies conducted in Ethiopia on adults yielded MOE values ranging from 404.2 to 1054.5 [38] and from 2818.9 to 6419.5 [42]. These findings indicate that exposure to AFM1 may increase the risk of developing HCC and represents a significant public health concern.

Previous risk characterization studies have shown that milk with AFM1 concentrations lower than 10 ng/kg should be included in the diet of young children to ensure safety, that is, to meet safety limits, i.e., HI below 1 and MOE above 10,000 [18,36].

## 4. Conclusions

This study found that the average incidence of AFM1 exceeding the ML in Croatia ranged from 5.67% to over 15%. The highest detection frequency was observed during the autumn of 2024. The low occurrence of AFM1 in milk (81.3% below LOD) is likely attributable to significantly improved preventive measures, control activities, and risk management procedures implemented during the harvesting, processing, and storage of cattle feed. However, the climate changes affecting Croatia, such as tropical-like heat stress and prolonged droughts during the summer months, along with alterations in precipitation patterns, create favorable conditions for mold growth and aflatoxin production.

In this research, the first assessment of the risk of exposure to AFM1 was conducted for different age groups and both genders. The EDI values for females and males were slightly different, with females exhibiting marginally higher values than males across all age groups, except for children and adolescents. The risk characterization, based on HI and MOE evaluation, indicated that toddlers and children face the highest risks. This can be attributed to their lower body weights and higher milk consumption. The risk assessment indicated that milk consumption does not present a significant health risk associated with AFM1 exposure for adults and the elderly. However, individuals who consume large quantities of milk across all age groups are exposed to elevated concentrations of AFM1, which poses a potential threat to public health.

In conclusion, it is essential to continue efforts aimed at reducing feed contamination with AFB1 and, consequently, lowering the concentration of AFM1 in milk. This will help mitigate the risk of exposure to AFM1 among milk consumers, particularly children of all ages. An effective control system for AFM1 in milk is of paramount importance. The application of the Italian strategy of introducing an “attention limit” for feed and milk within national control plans, especially during extreme weather conditions, could serve as a viable solution to achieve even better results.

## Figures and Tables

**Figure 1 foods-14-02396-f001:**
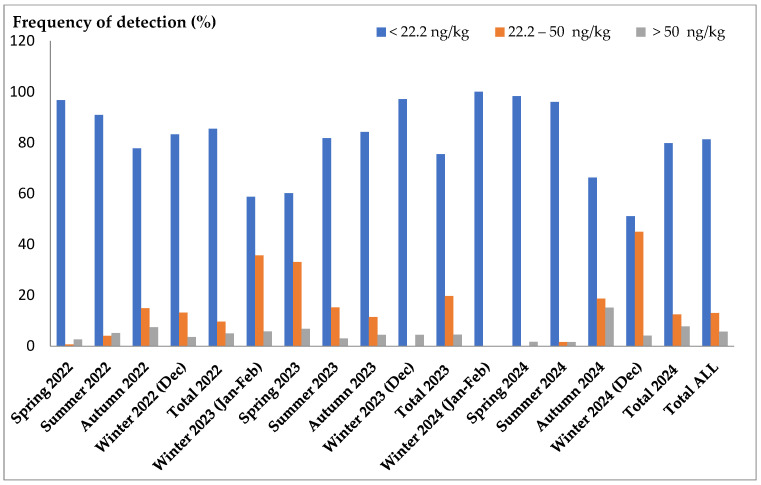
Frequencies of AFM1 detection in milk over the three-year period within three concentration ranges: below the limit of detection (LOD, 22.2 ng/kg), in the range from the LOD to the maximum level (ML, 50 ng/kg), and above the ML.

**Table 1 foods-14-02396-t001:** Mean concentrations of AFM1 in raw cow milk collected in Croatia over a three-year period (2022–2024).

Season	Total N	Mean ± SD (ng/kg)		AFM1 ≥ 50 ng/kg	
		Total	22.2–50	Mean ± SD (ng/kg)	Range (ng/kg)
Spring 2022	151	17.1 ± 42.1	39.4	230.0 ± 140.2	52.5–434.1
Summer 2022	175	16.5 ± 23.4	27.6 ± 4.48	103.1 ± 50.1	61.5–231.0
Autumn 2022	269	19.8 ± 23.7	32.8 ± 8.09	83.7 ± 46.7	51.3–269.8
Winter 2022 (Dec)	250	20.6 ± 79.2	29.6 ± 6.60	206.0 ± 369.4	50.4–1250
*Total 2022*	*845*	*18.8 ± 49.6*	*31.1 ± 7.48*	*129.9 ± 191.7*	*50.4–1250*
Winter 2023 (Jan–Feb)	87	21.4 ± 18.3	27.7 ± 5.16	84.5 ± 14.9	62.6–99.7
Spring 2023	88	21.2 ± 16.5	29.8 ± 4.98	69.9 ± 13.0	55.3–92.0
Summer 2023	164	16.4 ± 14.7	31.2 ± 5.45	84.1 ± 22.1	57.9–124.5
Autumn 2023	114	16.7 ± 18.1	31.0 ± 7.30	87.8 ± 32.6	53.5–129.4
Winter 2023 (Dec)	35	14.2 ± 18.6	nd	121.1	121.1
*Total 2023*	*488*	*18.1 ± 16.9*	*29.7 ± 5.73*	*82.8 ± 23.8*	*53.5–129.4*
Winter 2024 (Jan–Feb)	46	11.1 ± 0.02	nd	nd	nd
Spring 2024	120	12.6 ± 12.4	nd	103.8 ± 25.6	77.7–129.8
Summer 2024	125	13.1 ± 10.9	40.9 ± 1.73	89.4 ± 10.6	78.7–100.0
Autumn 2024	278	28.9 ± 54.0	30.9 ± 8.34	104.5 ± 109.6	50.8–742.0
Winter 2024 (Dec)	49	20.0 ± 11.3	26.6 ± 3.19	57.7 ± 7.42	50.3–65.1
*Total 2024*	*618*	20.5 ± 37.8	*30.0 ± 7.64*	*101.9 ± 103.1*	*50.3–742.0*
*TOTAL ALL*	*1951*	*19.2 ± 39.9*	*30.2 ± 6.95*	*108.5 ± 136.4*	*50.3–1250*
		nd—not detected			

**Table 2 foods-14-02396-t002:** Estimated daily intake (EDI) and risk assessment of AFM1 exposure for different age-sex groups in the Croatian population through milk consumption during the three-year period from 2022 to 2024.

Age Group/Years	Mean /P95	Milk Consumption (g/kg bw/day)		Total				22.2–50 ng/kg				>50 ng/kg			
				EDI ^a^		HI ^b^		EDI		HI		EDI		HI	
		Female	Male	Female	Male	Female	Male	Female	Male	Female	Male	Female	Male	Female	Male
Toddlers 1–3	Mean	22.21	20.05	0.42	0.38	2.13	1.92	0.67	0.61	3.35	3.03	2.41	2.18	12.1	10.9
	P95 ^c,d,e^	51.52	45.60	2.84	2.52	14.2	12.6	2.28	2.02	11.4	10.1	12.7	11.3	63.6	56.3
Children 3–9	Mean	13.42	14.09	0.26	0.27	1.29	1.35	0.41	0.43	2.03	2.13	1.46	1.53	7.28	7.64
	P95	29.18	30.15	1.61	1.67	8.07	8.33	1.29	1.33	6.45	6.67	7.20	7.44	37.2	36.0
Adolescents 10–17	Mean	4.38	4.86	0.084	0.093	0.42	0.47	0.13	0.15	0.66	0.73	0.48	0.53	2.38	2.64
	P95	10.33	11.97	0.57	0.66	2.85	3.31	0.45	0.53	2.28	2.65	2.55	2.96	12.8	14.8
Adults 18–65	Mean	2.78	2.40	0.053	0.046	0.27	0.23	0.084	0.072	0.42	0.36	0.30	0.26	1.51	1.30
	P95	6.60	5.74	0.36	0.32	1.82	1.59	0.29	0.25	1.46	1.27	1.63	1.42	8.15	7.09
Eldery 66–74	Mean	2.46	1.92	0.047	0.037	0.24	0.18	0.074	0.058	0.37	0.29	0.27	0.21	1.33	1.04
	P95	6.09	5.32	0.34	0.29	1.68	1.47	0.27	0.24	1.35	1.18	1.50	1.34	7.52	6.57
Very eldery >75	Mean	2.63	2.29	0.050	0.044	0.25	0.22	0.079	0.069	0.40	0.35	0.29	0.25	1.43	1.24
	P95	6.40	5.71	0.35	0.31	1.77	1.58	0.28	0.25	1.41	1.26	1.58	1.41	7.90	7.05

^a^ Estimated daily intake, EDI (ng/kg bw/day); ^b^ HI, Hazard index; ^c^ P95 for total group = 55.3 ng/kg; ^d^ P95 for group 22.2–50 ng/kg = 44.2 ng/kg; ^e^ P95 for group > 50 ng/kg = 247.0 ng/kg.

**Table 3 foods-14-02396-t003:** Risk evaluation using margin of exposure (MOE) approach.

		MOE					
Age Group/Years	Mean /P95	Total		22.2–50 ng/kg		>50 ng/kg	
		Female	Male	Female	Male	Female	Male
Toddlers 1–3	Mean	9523	10,526	5970	6557	1659	1834
	P95	1408	1587	1754	1980	314.9	354.0
Children 3–9	Mean	15,384	14,814	9756	9302	2740	2614
	P95	2484	2395	3100	3007	556	538
Adolescents 10–17	Mean	47,619	43,011	30,769	26,667	8333	7547
	P95	7018	6060	8889	7547	1569	1351
Adults 18–65	Mean	75,471	86,956	47,619	55,556	13,333	15,384
	P95	11,111	12,500	13,793	16,000	2454	2817
Eldery 66–74	Mean	85,106	108,108	54,054	68,965	14,815	19,048
	P95	11,764	13,793	14,814	16,667	2667	2985
Very eldery >75	Mean	80,000	90,909	50,633	57,971	13,793	16,000
	P95	11,428	12,903	14,286	16,000	2532	2837

## Data Availability

The original contributions presented in the study are included in the article. Further inquiries can be directed to the corresponding author.

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
