# Peer review of "Occurrence of Aflatoxin M1 over Three Years in Raw Milk from Croatia: Exposure Assessment and Risk Characterization in Consumers of Different Ages and Genders"

_foods, 2025, doi:10.3390/foods14132396_

Round 1
Reviewer 1 Report
Comments and Suggestions for Authors
The topic of this manuscript is interesting. It is important to conduct research on the risk assessment of AFM1 in milk in Croatia. However, it needs revision to meet the journal standard. The details comments in as follows:
Title
- P1 Line 2-4. Please add the specific types of milk, raw milk, pasteurized milk, or UHT milk.
Abstract
- P1 Line 16-17. It should be better to and the background in the beginning of abstract. And please provide detailed sampling information, including sample quantity, sample type, and sampling season.
- P1 Line 17-19. Why were the target hazard quotient (THQ) and hazard index (HI) indicators ignored, which are also important for the risk assessment? In addition, please add the specific age group.
- P1 Line 19-20. Please add the mean concentration of AFM1. Additionally, it should be “maximum residue levels (MRL)” instead of “maximum levels (ML)”.
- What is the comparison result between males and females?
- P1 Line 30. Please add a conclusion for the present study.
Introduction
- P2 Line 63-77. Please support studies on varying levels of AFM1 in milk across different seasons in the same region.
- The introduction lacks explanation on conducting AFM1 hazard analysis for different age groups and genders, please add some related references.
Materials and Methods
- P3 Line 85-86. Please add specific sampling information, including the number of samples for each year and season.
- P4 Line 134-155. Please add data on weight and milk consumption for different age groups required to calculate the risk assessment indicators.
Results and Discussion
- P8 Line 294-295. Generally speaking, values below LOQ are usually set to 0 or use LOQ half to calculate. In this study, specific detection values are used. Is this value accurate and reliable, and is there any relevant literature to support it?
- P9 Line 327-329. Table 1, due to the large detection range of AFM1, please add the median value in the results of different seasons.
Conclusions
- P12 Line 461-489. The conclusion section is too cumbersome, please condense the important content for summary.
Author Response
The topic of this manuscript is interesting. It is important to conduct research on the risk assessment of AFM1 in milk in Croatia. However, it needs revision to meet the journal standard. The details comments in as follows:
Title
- P1 Line 2-4. Please add the specific types of milk, raw milk, pasteurized milk, or UHT milk.
Title: corrected to: raw milk
Abstract
- P1 Line 16-17. It should be better to and the background in the beginning of abstract. And please provide detailed sampling information, including sample quantity, sample type, and sampling season.
- P1 Line 16-17: sentence is corrected to: In this study, the frequency of aflatoxin M1 (AFM1) occurrence in raw milk was investigated across different seasons over a three-year period from 2022 to 2024 in Croatia.
- Explanation: The authors believe that this order of facts in the abstract is more useful to the reader than putting sentences from the end of the abstract at the beginning. Listing more facts in the abstract would drastically increase the word count.
- P1 Line 17-19. Why were the target hazard quotient (THQ) and hazard index (HI) indicators ignored, which are also important for the risk assessment? In addition, please add the specific age group.
- P1 Line 27-29: HI is added in sentence: “The risk assessment, based on the HI and MOE indicated that toddlers and children are at the highest risk of exposure to AFM1, which raises significant health concerns.“
The risk assessment based on the HI indicated...”
- P1 Line 18-19: listing all age groups would again increase the word count of the abstract, so it is better to correct from “all consumer age groups“ to „different age groups“. Sentence is also corrected according to other reviwer comment to: “Risk assessment was conducted using estimated daily intake (EDI), hazard index (HI), and margin of exposure (MOE) for various age groups and both genders.”
- P1 Line 19-20. Please add the mean concentration of AFM1. Additionally, it should be “maximum residue levels (MRL)” instead of “maximum levels (ML)”.
- P1 Line 23: sentence is added: “The average mean value of AFM1 over the three-year period was 19.2 ng/kg.“
- Explanation: maximum level (ML) is a new terminology due to Commission Regulation (EU) 2023/915
- What is the comparison result between males and females?
- P1 Line 26: sentence is added: „The EDI values for females and males were slightly different.“
- P1 Line 30. Please add a conclusion for the present study.
- Explanation: We believe that the last three sentences are sufficient for the conclusions of this study. In this form, with correction that reviwer asked the abstract already have 219 words, so adding more sentences would significantly increase the word count.
Introduction
- P2 Line 63-77. Please support studies on varying levels of AFM1 in milk across different seasons in the same region.
- P2 Line 71-76: sentences was added: “AFM1 levels fluctuate across various seasons and years of observation. High frequencies of AFM1 occurrence exceeding the ML were recorded during the winter and spring months of 2013 in Croatia [24], while lower frequencies were observed during the five-year period from winter 2016 to winter 2022 [21]. A study conducted in Serbia from 2015 to 2018 showed variations in AFM1 levels, with high frequencies of occurrence noted in winter and autumn [30].”
- The introduction lacks explanation on conducting AFM1 hazard analysis for different age groups and genders, please add some related references.
- P2 Line 82-87: sentences are added: “Therefore, the risks associated with exposure to AFM1, as well as the characterization of these risks across different age groups and genders, have been examined in various studies. These studies focus on calculating the Estimated Daily Intake (EDI) and the Hazard Index (HI) [2,6,14,18,21]. Furthermore, for genotoxic and carcinogenic compounds such as aflatoxins, risk assessment using the Margin of Exposure (MOE) approach has been recommended [5] and implemented [1,3,23].“
Materials and Methods
- P3 Line 85-86. Please add specific sampling information, including the number of samples for each year and season.
- P3 Line 97-99: sentence is added: The total number of samples collected during the spring, summer, autumn, and winter seasons over three years is as follows: in 2022: 151, 175, 269 and 250; in 2023: 87, 88, 164 and 114; in 2024: 120, 125, 278 and 49.
- P4 Line 134-155. Please add data on weight and milk consumption for different age groups required to calculate the risk assessment indicators.
- P3 Line 158-159: Given that this is a large amount of data for milk consumption for different age groups with values expressed as mean and P95, it is noted in parentheses that these values are listed in Table 2 in the milk consumption column: (values are presented in the milk consumption column of Table 2).
Results and Discussion
- P8 Line 294-295. Generally speaking, values below LOQ are usually set to 0 or use LOQ half to calculate. In this study, specific detection values are used. Is this value accurate and reliable, and is there any relevant literature to support it?
- Explanation: we agree with the remark. Therefore, we put total results by the seasons in the first column of Table 1, whith LOD/2 values for aflatoxin M1 values below the LOD.
- P4 Line 180-181: Sentence is add: “Concentrations below the LOD were set to half of the LOD values.”
- P5 Line 186-188: Sentence was corrected to: “The mean concentrations of AFM1 were determined for the total number of samples collected per season over three years, as well as for two concentration ranges: between the LOD and the ML, and above the ML.”
- P5 Line 190-191: Sentence corrected to: „The total mean concentrations across the seasons over three years ranged from 11.1 to 28.9 ng/kg, with an overall mean value of 19.2 ng/kg.”
- P9 Line 327-329. Table 1, due to the large detection range of AFM1, please add the median value in the results of different seasons.
- Explanation: thanks for the suggestion, however we think that by adding columns for total mean values per seasons we have a real picture of afla M1 concentrations in the seasons over these three years. Also, the results of the study were compared with the mean values in other studies (not with median values) so the median values would not be commented on.
Conclusions
- P12 Line 461-489. The conclusion section is too cumbersome, please condense the important content for summary.
P12 Line 486-492: unnecessary sentences have been deleted and concluding sentences have been added: ” The risk characterization, based on HI and MOE evaluation indicated that toddlers and children face the highest risks. This can be attributed to their lower body weights and higher milk consumption. The risk assessment indicated that milk consumption does not present a significant health risk associated with AFM1 exposure for adults and the elderly. However, individuals who consume large quantities of milk across all age groups are exposed to elevated concentrations of AFM1, which poses a potential threat to public health.”

Reviewer 2 Report
Comments and Suggestions for Authors
This manuscript studies the frequency of aflatoxin M1 (AFM1) occurrence over a three-year period from 2022 to 2024 in Croatia, which is very interesting. However, there are some shortcomings in this paper.
1. This manuscript evaluates all consumer age groups, including infants. The question is: would infants be the consumer group of milk produced in the current year? Breast milk or milk powder should be food for infants. Milk powder factories should not use milk with excessive aflatoxin M1 as raw materials, even if milk powder is processed from milk produced in the current year. The same question may also apply to toddlers. If this premise is true, please revise the relevant content of the paper.
2. It is recommended to change the order of 3.2 “AFM1 concentration levels. Seasonal frequency of AFM1 occurrence” and 3.1 “Seasonal frequency of AFM1 occurrence”.
3. Line 141. It is recommended to change 49.9 ng/kg to 50 ng/kg. The same applies to Figure 1, Table 1 and Table 2.
4. Line 291-292. It is recommended to use the abbreviations for LOD and ML here.
5. Table 1. How did you get the data below LOD (<22.2)? How do you calculate the average of these data below LOD? It is recommended to add the number of test samples with different ranges of data. The number of significant figures should be consistent.
6. Line461-463. “This study determined that the average incidence of AFM1 above the maximum limit (ML) value in Croatia ranged from 5.67% to an increased detection frequency of over 15%, observed only during the autumn of 2024. ” It is recommended to revise this sentence carefully to avoid ambiguity. It is recommended to use the abbreviations for ML in this sentence.
Author Response
This manuscript studies the frequency of aflatoxin M1 (AFM1) occurrence over a three-year period from 2022 to 2024 in Croatia, which is very interesting. However, there are some shortcomings in this paper.
This manuscript evaluates all consumer age groups, including infants. The question is: would infants be the consumer group of milk produced in the current year? Breast milk or milk powder should be food for infants. Milk powder factories should not use milk with excessive aflatoxin M1 as raw materials, even if milk powder is processed from milk produced in the current year. The same question may also apply to toddlers. If this premise is true, please revise the relevant content of the paper.
- Explanation: You are absolutely right about infants who use breast milk or formula containing powdered milk with a low percentage of cow's milk protein. Therefore, we exclude this group from the manuscript and Tables 2 and 3, but remain data for the toddlers and children who consume milk. Accordingly, we have removed all data refereed to the infant group from the results (Table 1) and discussion. Additionally, total results by the seasons was added in the first column of Table 1.
- Line 371-372: New sentence was added: “For toddlers and children, the highest EDI values for mean AFM1 ranged from 1.46 to 2.41 ng/kg bw/day for both females and males. These values were observed in results exceeding 50 ng/kg.”
- It is recommended to change the order of 3.2 “AFM1 concentration levels. Seasonal frequency of AFM1 occurrence” and 3.1 “Seasonal frequency of AFM1 occurrence”.
- Explanation: as recommended order of two chapter is changed to 3.1 “AFM1 concentration levels.” and 3.2 “Seasonal frequency of AFM1 occurrence”.
- References: The order of references from 40 to 58 has been corrected in text and references list.
- Line 141. It is recommended to change 49.9 ng/kg to 50 ng/kg. The same applies to Figure 1, Table 1 and Table 2.
- Line 152, Figure 1, Table 1, Table 2: 49.9 ng/kg changed to 50 ng/kg - Line 291-292. It is recommended to use the abbreviations for LOD and ML here.
- Line 309-310: corrected to LOD and ML - Table 1. How did you get the data below LOD (<22.2)? How do you calculate the average of these data below LOD? It is recommended to add the number of test samples with different ranges of data. The number of significant figures should be consistent.
- Explanation: we agree with the remark. Therefore, we put total results by the seasons in the first column of Table 1, whit LOD/2 values for aflatoxin M1 values below the LOD.
- Line 180-181: Sentence is add: “Concentrations below the LOD were set to half of the LOD values.”
- Line 186-188: Sentence was corrected to: “The mean concentrations of AFM1 were determined for the total number of samples collected per season over three years, as well as for two concentration ranges: between the LOD and the ML, and above the ML.”
- Line 190-191: Sentence corrected to: „The total mean concentrations across the seasons over three years ranged from 11.1 to 28.9 ng/kg, with an overall mean value of 19.2 ng/kg.”
- Line 461-463. “This study determined that the average incidence of AFM1 above the maximum limit (ML) value in Croatia ranged from 5.67% to an increased detection frequency of over 15%, observed only during the autumn of 2024. ” It is recommended to revise this sentence carefully to avoid ambiguity. It is recommended to use the abbreviations for ML in this sentence.
- Lines 473-475: sentence was corrected to: This study found that the average incidence of AFM1 exceeding the ML in Croatia ranged from 5.67% to over 15%. The highest detection frequency was observed during the autumn of 2024.

Reviewer 3 Report
Comments and Suggestions for Authors
The study reveals the authors' experience in this research topic, having previously published other results. A few observations should be mentioned:
Line 137: Estimated Daily Intake (EDI) - please provide a bibliographic source for this calculation equation.
Since AFM1 is considered carcinogenic, as is known, there is no TDI based on a dose of no observable effect. In this situation, to characterize the risk of oral aflatoxin exposure, why was the calculation of MOEs established by EFSA [ e.g. Schrenk D., Bignami M., Bodin L., Chipman J.K., del Mazo J., Grasl-Kraupp B., Hogstrand C., Hoogenboom L.R., Leblanc J.C., Nebbia C.S., et al. (EFSA Panel on Contaminants in the Food Chain) Scientific opinion—Risk assessment of aflatoxins in food. EFSA J. 2020;18:112. doi: 10.2903/j.efsa.2020.6040] not addressed?
A well-conducted and presented study.
Author Response
The study reveals the authors' experience in this research topic, having previously published other results. A few observations should be mentioned:
Line 137: Estimated Daily Intake (EDI) - please provide a bibliographic source for this calculation equation.
- Line 150: reference is added: [21]
Since AFM1 is considered carcinogenic, as is known, there is no TDI based on a dose of no observable effect. In this situation, to characterize the risk of oral aflatoxin exposure, why was the calculation of MOEs established by EFSA [ e.g. Schrenk D., Bignami M., Bodin L., Chipman J.K., del Mazo J., Grasl-Kraupp B., Hogstrand C., Hoogenboom L.R., Leblanc J.C., Nebbia C.S., et al. (EFSA Panel on Contaminants in the Food Chain) Scientific opinion—Risk assessment of aflatoxins in food. EFSA J. 2020;18:112. doi: 10.2903/j.efsa.2020.6040] not addressed?
- Line 82-87: in Introduction chapter was added: “Therefore, the risks associated with exposure to AFM1, as well as the characterization of these risks across different age groups and genders, have been examined in various studies. These studies focus on calculating the Estimated Daily Intake (EDI) and the Hazard Index (HI) [2,6,14,18,21]. Furthermore, for genotoxic and carcinogenic compounds such as aflatoxins, risk assessment using the Margin of Exposure (MOE) approach has been recommended [5] and implemented [1,3,23].“
- Lines 167-177: paraghraph under chapter 2.5. Dietary Exposure and Risk Characterization about MOE calculation was added: “Risk characterization was also conducted using the MOE which serves as an index for assessing the risk of oral exposure to carcinogenic and genotoxic substances, such as AFM1 [5]. The European Food Safety Authority (EFSA) establishes the MOE calculation as follows: MOE = BMDL10/EDI, where BMDL10 refers to the benchmark dose level confidence limit of 10% (BMDL10). The BMDL10 ratio was established at 400 ng/kg body weight per day for the induction of hepatocellular carcinomas (HCC) in male rats following exposure to AFB1, but in the absence of a specific BMDL10 for AFM1 a potency factor of 0.1 relative to AFB1 was utilized. For MOE values equal to or greater than 10,000, it can be concluded that milk AFM1 poses minimal concern regarding public health. However, when the MOE is less than 10,000, it presents a significant risk of HCC for consumers [1,3,5,37].”
- Lines 438-467: results in Table 3 for MOE and interpretation about MOE is added: “Further risk characterization of AFM1 exposure was conducted using MOE approach which is particularly important as it assesses the carcinogenicity of AFM1 to the liver. The results of the MOE assessment are presented in Table 3. The MOE values, determined based on the mean values of the total results, show values below 10,000 only for female toddlers (9,523), while the values for males and other age groups exceed this safety limit. However, for the other two groups of results, the group between the LOD and ML, and those above the ML, MOE values fall below the safe limit for both female and male toddlers (ranging from 1,659 to 6,557) and for children (ranging from 2,614 to 9,756). This indicates that these two age groups are at risk due to the consumption of raw cow milk, raising significant public health concerns.
Furthermore, the MOE assessment conducted for EDI of P95 values revealed values below 10,000 for three groups of children: toddlers, children, and adolescents (ranging from 314.0 to 8,889). In addition, for all three age groups of adults in the AFM1 score group above ML, MOE values below the safe limit were also observed. MOE values exceeding the safe limit were calculated for three adult groups, considering both the total group of results and those results that fell between LOD and the ML (ranging from 11,111 to 16,667).
The values obtained in this study are consistent with EFSA's evaluation of MOEs for different age groups and support the conclusion that the results indicate a health concern for younger populations [5]. In an Italian study conducted between 2014 and 2020, MOE values were calculated for medium and high consumers [48]. As in this study, values below the safe limit were identified for toddlers, while for the children's group, they were below the safe limit only for high consumers. For other age groups, MOE values exceeded 10,000. However, a recent study from China reported MOE values exceeding 10,000, even for children aged 2 to 4 years; therefore, there were no public health concerns [36]. Conversely, studies conducted in several countries indicate an increased risk of exposure to AFM1 for older age groups. A study conducted in Serbia and Greece in 2018 found MOE values below the safe limit, thereby indicating an increased health risk for students aged 18 to 27 years [40]. Additionally, studies from Ghana reported MOE values below 10,000 for infants, toddlers, children, adolescents (ranging from 1,600 to 5,000), and adults (ranging from 2,105 to 6,666) [1]. Furthermore, studies conducted in Ethiopia on adults yielded MOE values ranging from 404.2 to 1,054.5 [38] and from 2,818.9 to 6,419.5 [42]. These findings indicate that exposure to AFM1 may increase the risk of developing HCC and represents a significant public health concern.”
- Lines 468-470; sentence is modified to: “Previous risk characterization studies have shown that milk with AFM1 concentrations lower than 10 ng/kg should be included in the diet of young children to ensure safety, i.e. to meet safety limits, i.e. HI below 1 and MOE above 10,000 [18,36].”
- Reference 5. is corrected to: EFSA Panel on Contaminants in the Food Chain (CONTAM), Schrenk, D.; Bignami, M.; Bodin, L.; Chipman, J.K.; del Mazo, J.; Grasl-Kraupp, B.; Hogstrand, C.; Hoogenboom, L.R.; Leblanc, J.C.; Nebbia, C.S.; et al. Scientific opinion—Risk assessment of aflatoxins in food. EFSA J. 2020, 18(3), 6040.
A well-conducted and presented study.
